# Etiology of acute febrile illnesses in Southern China: Findings from a two-year sentinel surveillance project, 2017–2019

**Jeanette J. Rainey**[1], **Casey Siesel**[2], **Xiafang Guo**[3], **Lina Yi**[4], **Yuzhi Zhang**[1], **Shuyu Wu**[1], **Adam L. Cohen**[2], **Jie Liu**[5,6], **Eric Houpt**[5], **Barry Fields**[2], **Zhonghua Yang**[3]*, **Changwen Ke**[4]*

1 Division of Global Health Protection, United States Centers for Disease Control and Prevention, Beijing, China, 2 Division of Global Health Protection, United States Centers for Disease Control and Prevention, Atlanta, Georgia, United States of America, 3 Yunnan Institute of Parasitic Diseases, Pu'er, Yunnan, China, 4 Center for Disease Control, Guangzhou, Guangdong, China, 5 Division of Infectious Diseases and International Health, Department of Medicine, University of Virginia, Charlottesville, Virginia, United States of America, 6 School of Public Health, Qingdao University, Qingdao, Shandong, China

* kechangwen@cdcp.org.cn (CK); 18987903669@163.com (ZY)

## Abstract

### Background

Southern China is at risk for arborvirus disease transmission, including Zika virus and dengue. Patients often present to clinical care with non-specific acute febrile illnesses (AFI). To better describe the etiology of AFI, we implemented a two-year AFI surveillance project at five sentinel hospitals in Yunnan and Guangdong Provinces.

### Methods

Between June 2017 and August 2019, we enrolled patients between 2 and 65 years of age presenting at one sentinel hospital in Mengla County, Yunnan, and four in Jiangmen City, Guangdong, with symptoms of AFI (acute onset of fever $\geq$ 37.5°C within the past 7 days) without respiratory symptoms or diarrhea. Demographic, epidemiologic, and clinical information was obtained and entered into a web-based AFI surveillance database. A custom TaqMan Array card (TAC) was used to test patients' whole blood specimens for 27 different pathogens using real-time polymerase chain reaction assays.

### Results

During the two-year project period, 836 patients were enrolled; 443 patients from Mengla County and 393 patients from Jiangmen City. The median age was 33 years [range: 2–65], and most were hospitalized [641, 77%]. Of 796 patients with valid TAC results, 341 (43%) were positive for at least one of the 10 unique pathogens detected. This included 205 (26%) patients positive for dengue virus, 60 (8%) for *Orientia tsutsugamushi*, and 42 (5%) for *Coxiella burnetii*. Ten patients (1%) in Jiangmen City tested positive for malaria, 8 of whom reported recent travel outside of China. TAC results were negative for 455 (57%) patients. None of the patients had a positive TAC detection for Zika virus.

**Data Availability Statement:** The authors of the manuscript do not have the legal authority to share the data with our submission of the manuscript. The data collected and analyzed as part of this

project is covered by the Data Sharing agreement established at the beginning of the project, consistent with the rules and regulations of the China Health Commission. Access to the de-identified dataset can be requested in a letter/email to Dr. Ke Changwen at Kechangwen@cdcp.org.con (for the AFI case data from Jiangmen City) and Dr. Zhonghua Yang at 18987903669 (for the AFI case and control data from Mengla County).

**Funding:** The AFI sentinel surveillance project was funded by the U.S.-China Collaborative Program on Emerging and Re-emerging Infectious Diseases Cooperative Agreement #5U2GGH000961-04 The funders had no role in study design, data collection and analysis, decision to publish, or preparation of the manuscript.

**Competing interests:** The authors have declared that no competing interests exist.

## Conclusions

The project detected variability in the etiology of AFI in Southern China and highlighted the importance of differential diagnosis. Dengue, *O. tsutsugamushi*, and *C. burnetii* were the most frequently identified pathogens among enrolled AFI patients. As a non-notifiable disease, the frequent detection of *C. burnetii* is noteworthy and warrants additional investigation. The project provided a framework for routine surveillance for persons presenting with AFI.

## Introduction

Despite global improvements in access to health care and reduction in malaria transmission, the global burden of acute febrile illnesses (AFI) remains high [1–3]. Recent studies have suggested that almost 80% of patients presenting with acute febrile illness (AFI), including those in malaria endemic regions, are due to non-malaria etiologies [4]. Zika, dengue, and chikungunya are viral infections that often cause acute febrile illnesses and can be under- or misdiagnosed due to the lack of appropriate diagnostic testing [5–7]. As a result, many of these AFI patients are treated according to clinical presentation [8]. The reliance on clinical presentation alone can lead to the misuse of antimicrobials [9] as well as poor patient outcomes. This is particularly the case for Zika virus, which like many other causes of AFI, is difficult to diagnose [10] and may clinically mimic other infections such as *Salmonella enterica*, *Listeria* spp., and *Brucella* spp. [2, 3].

China is at risk for arbovirus transmission and AFI, primarily due to the climate, presence of mosquito vectors, and highly mobile populations. A recent scoping review identified two publications on AFI surveillance from China [11]. These publications included a multi-country investigation of typhoid fever from 2008 [12] and a surveillance project to identify persons infected with the novel Severe Fever with Thrombocytopenia Syndrome (SFTS) bunyavirus from 2011 [13]. Recently developed multi-pathogen molecular diagnostic platforms, such as the TaqMan Array Card (TAC) (Life Technologies, Carlsbad, CA), can be used to detect multiple targets (i.e., pathogens) from a single patient specimen [14–16]. These diagnostic platforms could help identify etiologies for various syndromes more efficiently than single pathogen specific assays, commonly used in older surveillance projects.

Although extensive research has been conducted elsewhere in Asia [17], information on the etiology of AFI in southern China remains limited. To address this knowledge gap, we implemented a two-year AFI sentinel surveillance project at five hospitals in Guangdong and Yunnan Provinces. Both provinces have experienced previous outbreaks of dengue and remain at risk for importation of vector-borne diseases [18–22]. We used the customized AFI TAC to identify the etiology of patients presenting with AFI and from a small number of healthy controls recruited from the same sentinel hospitals. The primary objectives of the project were to describe the geographic variability in the etiology of AFI, inform public health strategies to prevent and respond to emerging pathogens, and guide clinical management.

## Methods

### Study setting

The AFI surveillance project was conducted between July 2017 to June 2019 at four sentinel hospitals in Jiangmen City, Guangdong Province and at one sentinel hospital in Mengla

County, Yunnan Province [23]. Jiangmen City is in the Pearl River Delta in south-central Guangdong Province. In 2015, the city had a population of 3,850,000. Each year, over 300,000 overseas Chinese residents working in the Americas and Europe return to Jiangmen City to visit relatives or for trade, business, and tourism. Enping County is under the jurisdiction of Jiangmen City. The county reported 13 cases of Zika virus from March to April 2016, all imported from overseas Chinese residing in Venezuela [24]. *Aedes albopictus*–which is associated with dengue, chikungunya, and Zika virus transmission—is the predominant mosquito vector in Jiangmen City. Of the four sentinel hospitals in Jiangmen City, two were in Enping County and two in Xinhui District. Two were tertiary hospitals and two were township level facilities. All four provided inpatient care.

Mengla County is in southern Yunnan Province and shares borders with Vietnam, Laos, and Myanmar to the east, south, and west, respectively. The county had a population of 194,360 in 2015. Both *Aedes albopictus* and *Aedes aegypti* mosquitos are found in the county. The *A. aegypti* mosquito is the primary vector for several pathogens, including dengue and the Zika virus. Although malaria was declared officially eliminated from China in 2021 [25, 26], Mengla County continues to experience imported malaria cases from neighboring Asian countries, including Myanmar, where the *Anopheles* mosquito vector remains abundant [27]. The AFI project was implemented in a single tertiary hospital in Mengla County that provided inpatient care.

## Study design and case definitions

We defined AFI as a patient with fever (axillary temperature $\geq 37.5$˚C or history of fever) with onset within the last 7 days and no other targeted symptoms (e.g., cough, sore throat, diarrhea). Verbal reports of fever with onset within the last 7 days (e.g., temperature $\geq 37.5$˚C measured at home) were accepted for patients who were not febrile at the time of enrolment. All patients between 2 and 65 years of age meeting the AFI definition and seeking care at one of the five sentinel hospitals were eligible for enrolment.

Starting in July 2018, we initiated enrollment of controls, with the aim of establishing associations between pathogen detection and symptomatic illness [11]. Patients between 2 and 65 years of age seeking care for non-infectious illnesses, or persons accompanying patients who did not have an infectious illness and had not experienced an AFI within the last 30 days, were eligible as controls. The original goal was to enroll one control for every five cases. However, due to logistical challenges, including obtaining consent from the healthy adults, controls were only enrolled at Mengla County Hospital.

If eligibility criteria were met, each AFI patient and control was assigned a unique project ID and written informed consent was obtained. Project staff then captured demographic, clinical, epidemiologic risk (including recent travel), and vaccination history information. As part of routine care, AFI patients were tested for dengue using the Colloidal Gold NS1 antigen test (Wondfo, Guangzhou, China) when warranted and, in Mengla County, for malaria using the Wondfo One Step Malaria HRP2/pLDH (P.f/Pan) test cassette rapid diagnostic test (RDT, Wondfo). Staff recorded patient information, including RDT results when available, on standard paper investigation report forms. The same enrollment procedures and paper investigation report form were used for recruited controls. Clinical and RDT data were not captured from controls.

## TAC specimen collection and testing

Following completion of the investigation report form and routine care, project staff at the sentinel hospitals collected venous whole blood samples from AFI patients and enrolled

controls using a 5 ml EDTA vacutainer and transported the samples in a cold box (4˚C) to the project laboratory. Specimens were stored at -70˚C in designated laboratory in Jiangmen City and were periodically shipped on dry ice to the Guangdong Centers for Disease Control and Prevention in Guangzhou, China for diagnostic testing using TAC. The first half of the specimens stored at -70˚C in the designated laboratory in Mengla County were tested at the Entry-Exit Inspection and Quarantine Bureau in Shenzhen, Guangdong Province, and the remaining specimens were tested at the Guangdong Centers for Disease Control and Prevention in Guangzhou, China.

The TAC is a 384-well real-time PCR platform (i.e., 48 wells for each of the 8 samples) developed by Life Technologies (Carlsbad, CA). The AFI TAC layout included 13 viral, 11 bacterial, and 3 protozoan pathogens known to cause AFI. Based on prior validation work, duplicate wells were reserved for bacterial pathogens to improve the detection sensitivity because of their potential low pathogen load in blood, and single wells were reserved for others to maximize the ability of the card to detect a range of pathogens (Fig 1). The TAC testing methods have been previously described [14]. Briefly, total nucleic acid was extracted from up to 2.5 mL of whole blood using High Pure Viral Nucleic Acid (Roche Diagnostics) and eluted in 150 μL of elution buffer. 75μL of total nucleic acid extract was mixed with 25μL of TaqMan Fast Virus 1 Step Master mix (Thermo Fisher), in a 100μL reaction, then pipetted into the inlet port on the card. Cards were centrifuged (1 min at 1,200 rpm twice), sealed and the inlet ports were removed as directed by the manufacturer's instructions. All AFI TACs were run on the ViiA 7 or QuantStudio 7 Flex real-time PCR system (Thermo Fisher) using PCR cycling conditions comprising of 10 minutes at 50˚C, 20 seconds at 95˚C, followed by 45 two-step cycles of 3 seconds of 95˚C and 30 seconds at 60˚C. A positive result was defined by ≤35 cycle threshold (Ct) value. Bacteriophage MS2 and phocine herpesvirus (PhHV) were spiked to the blood samples during nucleic acid extraction as extrinsic controls to monitor extraction and amplification. One extraction blank was included in each batch of extraction to monitor lab contamination. The results were deemed valid only when the corresponding extrinsic controls or extraction blanks yielded valid results.

## Data management and analysis

Paper investigation forms were completed at the sentinel hospitals and maintained at Jiangmen City CDC and Mengla County CDC. On a weekly basis, sentinel hospital staff entered data from the investigation forms into the AFI Epidemiologic Dynamic Data Collection (EDDC) platform, a customized web-based surveillance interface tool and database [28]. The AFI EDDC platform was developed by China CDC with input from project staff at Guangdong CDC and Yunnan Institutes of Parasitic Diseases (YIPD) [23].

Data were downloaded from the AFI-EDDC database into Excel and imported into RStudio (R version 4.1.0, RStudio version 1.4.1717) for cleaning and analysis [29]. We described demographic characteristics, clinical presentation, epidemiologic risk factors and TAC results of enrolled AFI patients and performed bivariate and multivariate analyses to assess associations between demographic characteristics, epidemiologic risks, and TAC results. TAC results for enrolled controls were described. We calculated $X^2$ tests to examine differences in the demographic characteristics of AFI patients enrolled across the five sentinel hospitals, and generated odds ratios (OR) and adjusted odds ratios (aOR) from our bivariate and multivariate analyses, respectively. Due to frequent detection of dengue during the study period, we assessed the statistical agreement between TAC results and the NS1 dengue antigen RDT using Cohen's kappa statistic. We repeated this analysis to assess agreement between TAC results and the

| Port | |
|:---:|:---:|
| **Left** | **Right** |
| *Bartonella* | *Bartonella* |
| *Brucella* | *Brucella* |
| *Burkholderia pseudomallei* | *Burkholderia pseudomallei* |
| *Coxiella burnetti* | *Coxiella burnetti* |
| Chikungunya | CCHF |
| Bundibuygo & Sudan | Ebola |
| Dengue | Hepatitis E |
| *Leptospira* | *Leptospira* |
| Lassa | Mayaro |
| Nipah | Marburg & O'noyong-nyong |
| *Orientia tsutsugamushi* | *Orientia tsutsugamushi* |
| *Plasmodium* | *P. falciparum/vivax* |
| MS2 | PhHV |
| 18S | PhHV/MS2 |
| *Rickettsia* | Rift Valley Fever |
| *Salmonella* | *Rickettsia* |
| *Salmonella* Typhi | *Salmonella* |
| *Salmonella* Paratyphi A | *Salmonella* Typhi |
| *Streptococcus pneumoniae* | *Leishmania* |
| *Streptococcus suis* | *Streptococcus pneumoniae* |
| *T. brucei* | *Streptococcus suis* |
| *Yersina pestis* | Yellow Fever |
| Zika pan (800) | 16S |
| | Zika pan (1000) |

**Fig 1. Layout of AFI TaqMan Array Card (TAC), June 2017 –July 2019, China.**

malaria RDT used in Mengla County. A p-value of 0.05 was used to assess statistical significance.

## Ethical approval

This study was approved by the Institutional Review Boards at China CDC, Guangdong Provincial CDC, YIPD, and US CDC. Written consent was obtained from eligible patients, or the patient's legal guardian if the patient was < 18 years of age. Guangdong CDC and Yunnan Institutes of Parasitic Diseases developed and disseminated guidelines for counseling and care for any pregnant women identified with a positive Zika virus infection during the project.

## Results

### Enrollment demographics and clinical presentation

From 21 June 2017–22 August 2019, 836 patients were screened for eligibility and enrolled in the surveillance project. This included 393 patients enrolled at the four sentinel hospitals in Jiangmen City and 443 at the single sentinel hospital in Mengla County (Table 1). Of the enrolled patients, 187 (22%) were children and adolescents 2 to 17 years of age, and 649 (78%) were adults 18 to 65 years of age. More than half (471, 56%) of enrolled patients were male, and 263 (31%) were employed in farming, manufacturing, or fishing industries.

Most enrolled patients (809, 97%) were Chinese and 27 (3%) were foreign nationals, including 16 from Laos, 8 from Venezuela, and 1 each from Myanmar, South Africa, and Mexico. Across the two-year project period, AFI patient enrolment peaked in April and May ($X^2$ = 118.05, p-value < .001) in Jiangmen City and in September and October ($X^2$ = 241.49, p-value < .001) in Mengla County (Fig 2).

Thirty-five controls were screened for eligibility and enrolled at the sentinel hospital in Mengla County. Two (6%) controls were between 2 to 17 years of age and 33 (94%) were adults

**Table 1. Demographic characteristics of patients enrolled in the acute febrile illness surveillance project by location in China, June 2017 –August 2019.**

|  | Total | Jiangmen City | Mengla County | P-value |
|---|---|---|---|---|
|  | No. 836 | No. 393 | No. 443 |  |
| **Sex** |  |  |  | < .001 |
| Female | 365 (44%) | 146 (37%) | 219 (49%) |  |
| Male | 471 (56%) | 247 (63%) | 224 (51%) |  |
| **Age*** |  |  |  | .002 |
| Mean (SD) | 33 (±18) | 35 (±18) | 31 (±17) |  |
| Median (IQR) | 33 (21–48) | 34 (23–50) | 31 (17–46) |  |
| **Age Categories** |  |  |  | .063 |
| 2–5 | 53 (6%) | 24 (6%) | 29 (7%) |  |
| 5–17 | 134 (16%) | 51 (13%) | 83 (19%) |  |
| 18–45 | 403 (48%) | 189 (48%) | 214 (48%) |  |
| 46–65 | 246 (29%) | 129 (33%) | 117 (26%) |  |
| **Nationality** |  |  |  | .33 |
| Chinese | 809 (97%) | 383 (97%) | 426(96%) |  |
| Other** | 27 (3%) | 10 (3%) | 17 (4%) |  |
| **Occupation** |  |  |  | < .001 |
| Farmer, Manufacturing, Fisherman | 263 (31%) | 63 (16%) | 200 (45%) |  |
| Homemaker | 66 (8%) | 60 (15%) | 6 (1%) |  |
| Office worker | 47 (6%) | 29 (7%) | 18 (4%) |  |
| Other | 265 (32%) | 152 (39%) | 113 (26%) |  |
| Student | 185 (22%) | 81 (21%) | 104 (23%) |  |
| Transportation (taxi driver, truck driver) | 10 (1%) | 8 (2%) | 2 (0%) |  |
| **Education** |  |  |  | < .001 |
| Primary School | 185 (22%) | 64 (16%) | 121 (27%) |  |
| Lower than Primary | 18 (2%) | 3 (1%) | 15 (3%) |  |
| High School | 352 (42%) | 201 (51%) | 151 (34%) |  |
| Current Student | 171 (20%) | 69 (18%) | 102 (23%) |  |
| College | 110 (13%) | 56 (14%) | 54 (12%) |  |

*Age rounded down to full year.

**Other includes Laos (16), Venezuela (8), Mexico (1), Myanmar (1), and South Africa (1).

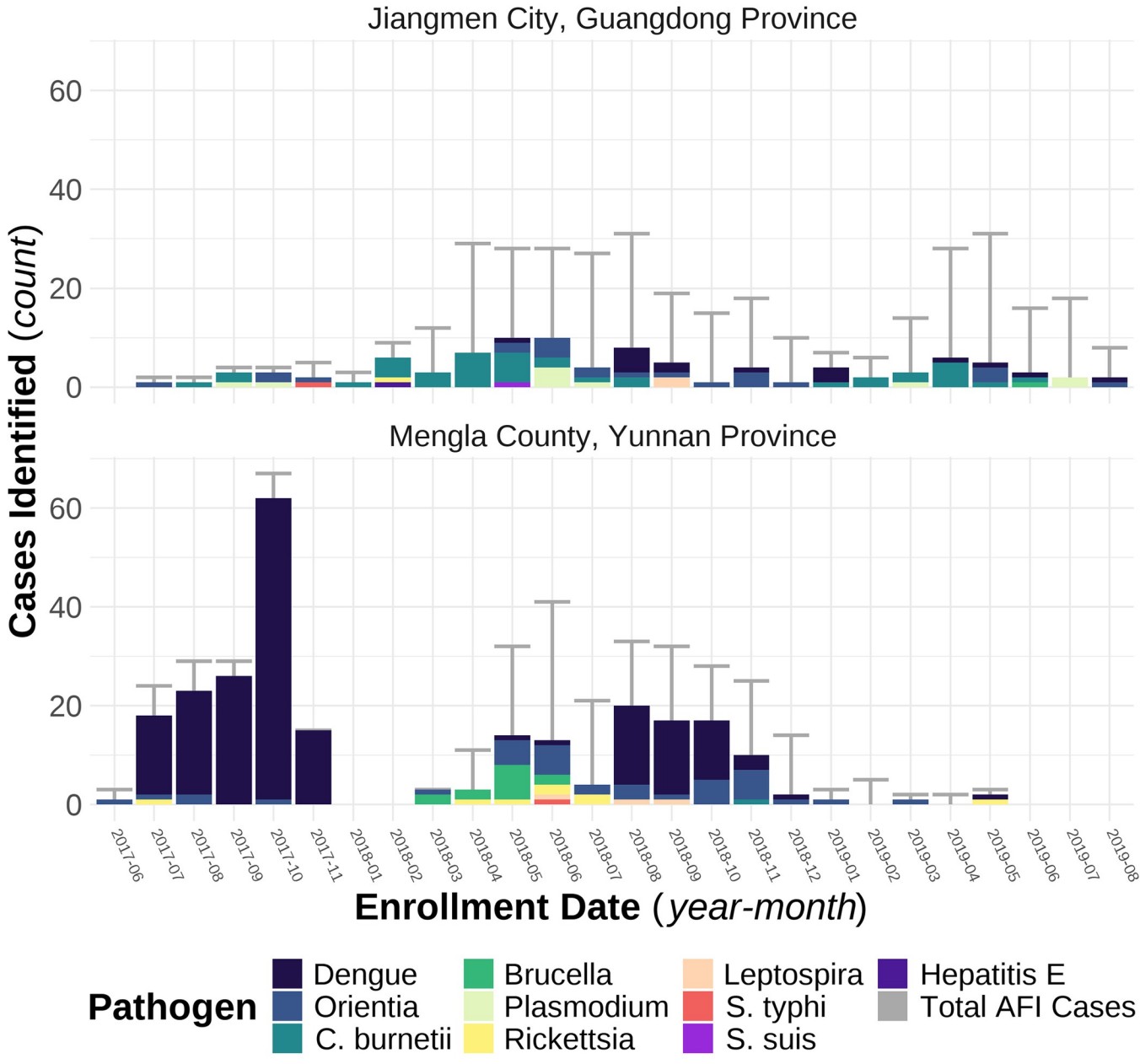

A temporal visualization of pathogen counts by province.
Cases with coinfections are treated as separate cases for each pathogen.

**Fig 2. Temporal distribution of patients enrolled in the acute febrile illness surveillance project by TaqMan Array Card (TAC) diagnostic test results and location, China, June 2017–July 2019.**

18 to 65 years of age. Approximately half (18, 51%) of the controls were male, and the majority (29, 83%) were employed in the farming, manufacturing, or fishing industry. Of the 35 controls, 26 (74%) were from Laos and 9 (26%) were Chinese nationals.

Of the 836 enrolled AFI patients, 825 (99%) presented with fever at the sentinel hospital (mean axillary temperature was 38.7°C [range: 37.5–41.8°C]). Enrolment relied on history of fever ≥ 37.5°C (within the last 7 days) for the remaining 11 (1%) patients. In addition to fever,

**Table 2. Clinical presentation\* of patients enrolled in the acute febrile illness surveillance project by location in China, June 2017 –July 2019.**

| Clinical Presentation | Total | Jiangmen City | Mengla County |
|---|---|---|---|
| | n = 836 | n = 393 | n = 443 |
| Fever at enrolment\*\* | 825 (99%) | 388 (99%) | 437 (99%) |
| Rash | 127 (15%) | 29 (7%) | 98 (22%) |
| Redeyes | 35 (4%) | 13 (3%) | 22 (5%) |
| Joint pain | 144 (17%) | 36 (9%) | 108 (24%) |
| Headache | 464 (56%) | 171 (44%) | 293 (66%) |
| Chills | 394 (47%) | 180 (46%) | 214 (48%) |
| Muscle Pain | 283 (34%) | 70 (18%) | 213 (48%) |
| Vomiting | 99 (12%) | 49 (12%) | 50 (11%) |
| Bloody Sputum | 9 (1%) | 4 (1%) | 5 (1%) |
| Bone Pain | 57 (7%) | 30 (8%) | 27 (6%) |
| Nose/Gum Bleeding | 14 (2%) | 7 (2%) | 7 (2%) |
| Swollen Joints | 10 (1%) | 4 (1%) | 6 (1%) |

\*Of the 836 AFI patients, 641 (77%) were hospitalized at the time of enrollment; 303 (77%) in Jiangmen City. and 338 (76%) in Mengla County. The remaining 195 (23%) patients were enrolled as outpatients.

\*\*Eleven (1%) patients were eligible based on reported history of fever within the last 7 days.

the most frequently reported AFI patient symptoms were headache (464, 56%), chills (394, 47%), and muscle pain (283, 34%) (Table 2). The majority of AFI patients (641, 77%) were hospitalized at the time of enrolment.

## Epidemiologic factors

Of the 836 enrolled patients, 70 (8%) reported travel within the three months prior to symptom onset—31 (44%) reported international travel and 39 (56%) reported travel to other provinces in China. One patient in Mengla County indicated receiving an organ transplant within 30 days of symptom onset, and 9 (1%) patients received blood or blood products during this same timeframe. Five patients were pregnant at the time of enrolment. None of the AFI patients reported previous contact with someone diagnosed with the Zika virus.

## TAC results

Of the 796 (95%) patients with valid TAC results, 333 (42%) patients were positive for a single pathogen and eight (<1%) patients had co-infections with two of the pathogens on TAC (Table 3, Table A in S1 File). Blood samples were not available for the remaining 40 (5%) patients. A total of 10 different bacterial, viral, and protozoan etiologies were detected using TAC. This included 10 unique pathogens detected from 95 patients in Jiangmen City and 7 unique pathogens from 246 patients in Mengla County. Positive TAC detections occurred primarily during April through June in Jiangmen City (n = 41/95, 43%) and August through October in Mengla County (n = 165/246, 67%).

The most common pathogens detected among enrolled patients in Mengla County were DENV (189; 45%), *Orientia tsutsugamushi* (37; 9%), and *Brucella* spp. (13; 3%). The most common pathogens in Jiangmen City were *Coxiella burnetii* (41, 11%), *O. tsutsugamushi* (23, 6%), and DENV (16, 4%). Of the 41 cases of *C. burnetii* in Jiangmen City, 29 (71%) were detected from patients enrolled at Xinhui District Hospital and another 10 (24%) from Enping County Hospital (Table B in S1 File). Ten (3%) patients in Jiangmen City had positive TAC detections

**Table 3. TaqMan Array Card (TAC) diagnostic test results* and cycle threshold values (Ct) for patients enrolled in the acute febrile illness surveillance project by location in China, June 2017–July 2019.**

| | Total | Median | Jiangmen City | Mengla County |
|---|---|---|---|---|
| | **n = 796** | **Ct Value (range)** | **n = 374** | **n = 422** |
| **Bacteria** | | | | |
| *Brucella* spp. | 14 (2%) | 34.1 (32.5–34.5) | 1 (0%) | 13 (3%) |
| *Coxiella burnetiid* | 42 (5%) | 30.0 (22.3–34.8) | 41 (11%) | 1 (0%) |
| *Leptospira* spp. | 5 (1%) | 32.5 (27.8–34.6) | 2 (1%) | 3 (1%) |
| *Orientia tsutsugamushi* | 60 (8%) | 30.5 (24.1–34.2) | 23 (6%) | 37 (9%) |
| *Rickettsia* spp. | 9 (1%) | 34.7 (32.4–34.9) | 1 (0%) | 8 (2%) |
| *Salmonella* Typhi | 2 (0%) | 34.6 (34.4–34.9) | 1 (0%) | 1 (0%) |
| *Streptococcus suis* | 1 (0%) | 21.4 | 1 (0%) | 0 (0%) |
| **Viruses** | | | | |
| Dengue virus | 205 (26%) | 23.9 (13.2–35.0) | 16 (4%) | 189 (45%) |
| Hepatitis E | 1 (0%) | 35.0 | 1 (0%) | 0 (0%) |
| **Protozoa** | | | | |
| *Plasmodium* spp. | 10 (1%) | 13.0 (5.56–34.7) | 10 (3%) | 0 (0%) |
| **No pathogen detected** | 455 (57%) | - | 279 (75%) | 176 (42%) |

*The number of TAC results is greater than the number of enrolled patients due to the eight patients who had co-infections with two of the pathogens on TAC.

for *Plasmodium* spp.. None of the AFI patients tested positive for Zika virus. TAC results were negative for all 27 pathogens for the 35 enrolled controls.

## Factors associated with positive TAC results

In our multivariate logistic regression model, reported residence in Mengla County (aOR = 5.5, p-value < .001) was the only factor associated with having a positive TAC detection, independent of other epidemiologic and demographic risk factors (Table C in S1 File). In our etiology specific analysis, TAC detections for dengue were more likely for patients enrolled in Mengla County (aOR = 30.7, p-value < .001), and less likely for males (aOR = 0.6, p-value = .020) and current students (aOR = .1, p-value = .001) (Table 4). TAC detections for *O. tsutsugamushi* were more likely among adults at least 18 years of age (aOR = 18.0, p-value < .001) (Table 5), while *C. burnetti* was more likely detected among males (aOR = 3.3, p-value = .005) and less likely among patients enrolled in Jiangmen City (aOR = 0.02, p-value < .001) (Table 6).

Additionally, nine of the ten patients with positive TAC results for *Plasmodium* spp. in Jiangmen City were male, and eight reported international travel within the three months prior to symptom onset, including to Venezuela, Kenya, Democratic Republic of the Congo, and Uganda. Dengue was the only pathogen detected among all age groups and both sexes. Two AFI patients with positive TAC results for dengue (in Mengla County and Jiangmen City) and one AFI patient with a positive TAC result for *C. burnetti* (in Jiangmen City) were pregnant at the time of enrolment.

## Clinical presentation of top six etiologies

The six most frequently detected AFI etiologies included DENV, *C. burnetti*, *O. tsutsugamushi*, *Brucella spp.*, *Rickettsia spp.*, and *Plasmodium spp.* Of the 340 positive TAC detections for one of these etiologies (among 333 patients), 229 (67%) reported a headache, 165 (49%) muscle pain, and 155 (46%) chills at the time of enrolment (Table D in S1 File). Most patients reporting a rash (70/81, 86%) and joint pain (65/89, 73%) tested positive for DENV. Eight (10%)

**Table 4. Association between patient characteristics/epidemiologic factors and detection of dengue on the TaqMan Array Card (TAC)\* diagnostic testing platform, China, June 2017–July 2019.**

| Characteristic/Epidemiologic risk | TAC+ Dengue | Crude OR (95% CI) | ORa (95% CI) |
|---|---|---|---|
| | n (%) | | |
| **Location** | | | |
| Mengla County | 189 (92.2%) | 18.1 (10.9–32.2) | 30.7 (16.8–60.7) |
| Jiangmen City | 16 (7.8%) | Ref | Ref |
| **Nationality** | | | |
| Other* | 4 (1.9) | 0.5 (0.1–1.4) | 0.3 (0.1–1.0) |
| Chinese | 201 (98.1%) | Ref | Ref |
| **Sex** | | | |
| Male | 92 (44.9%) | 0.5 (0.4–0.7) | 0.6 (0.4–0.9) |
| Female | 113 (55.1%) | Ref | Ref |
| **Age group** | | | |
| > = 18 years | 178 (86.8%) | 2.3 (1.5–3.7) | 0.7 (0.3–1.8) |
| 2–17 years | 27 (13.2%) | Ref | Ref |
| **Occupation** | | | |
| Farmer, manufacturing, fisherman | 87 (42.4%) | 3.6 (1.7–8.5) | 0.5 (0.2–1.4) |
| Transportation (e.g., taxi/bus driver) | 1 (0.5%) | 0.8 (0.1–5.5) | 0.4 (0.1–4.5) |
| Office worker | 15 (7.3%) | 3.4 (1.3–9.4) | 1.0 (0.3–4.0) |
| Student | 18 (8.8%) | 0.7 (0.3–1.9) | 0.1 (0.01–0.3) |
| Other | 76 (37.1%) | 2.8 (1.4–6.7) | 0.8 (0.3–2.2) |
| Homemaker | 8 (3.9%) | Ref | Ref |
| **Education** | | | |
| Less than primary school | 6 (2.9%) | 4.2 (1.3–12.6) | 0.8 (0.1–3.8) |
| Primary school | 55 (26.8%) | 3.3 (1.9–5.8) | 0.8 (0.2–2.8) |
| High school | 80 (39.0%) | 2.2 (1.3–3.8) | 0.7 (0.2–2.6) |
| College | 43 (21.0%) | 4.7 (2.6–8.7) | 1.4 (0.3–5.6) |
| Currently in school | 21 (10.2%) | Ref | Ref |
| **Travel prior to onset** | | | |
| No | 192 (93.7%) | 1.4 (0.7–2.7) | 0.5 (0.2–1.2) |
| Yes | 13 (6.3%) | Ref | Ref |

*Other includes Laos (16), Venezuela (8), Mexico (1), Myanmar (1), and South Africa (1).

other patients with a rash and 14 (16%) with joint pain tested positive for *O. tsutsugamushi*. This includes two patients in Mengla County with DENV-*O. tsutsugamushi* co-infections. Hospitalized- and out-patients had similar clinical presentations (headache, chills, and muscle aches) (all p-values > 0.10) at the time of enrolment.

## Agreement between TAC and RDTs

Rapid diagnostic tests for dengue and malaria were used at the sentinel hospitals as part of routine care. The overall statistical agreement between the NS1 antigen RDT and TAC results for dengue was high (Cohen's kappa statistic = 0.84, p-value < .001) (Table E in S1 File). Agreement was 0.85 in Mengla County and 0.009 in Jiangmen City. Of the 370 NS1 dengue negative cases in Jiangmen City, 16 were positive by TAC (corresponding Ct values ranged from 15.3–34.9), and of the two positive NS1 dengue positive cases, both were negative by TAC. None of the 10 AFI patients initially diagnosed with malaria in Mengla County using Wondfo Malaria RDT were confirmed with *Plasmodium* by TAC.

**Table 5. Association between patient characteristics/epidemiologic factors and detection of *Orientia tsutsugamushi* on the TaqMan Array Card (TAC)* diagnostic testing platform, China, June 2017–July 2019.**

| Characteristic/Epidemiologic risk | TAC+ *O. tsutsugamushi* | Crude OR (95% CI) | ORa (95% CI) |
|---|---|---|---|
| | n (%) | | |
| **Location** | | | |
| Mengla County | 37 (61.7%) | 1.4 (0.9–2.5) | 1.2 (0.6–2.2) |
| Jiangmen City | 23 (38.3%) | Ref | Ref |
| **Nationality** | | | |
| Other* | 2 (3.3%) | 1.0 (0.2–3.6) | 2.4 (0.3–11.8) |
| Chinese | 58 (96.7%) | Ref | Ref |
| **Gender** | | | |
| Male | 32 (53.3%) | 0.9 (0.5–1.5) | 0.9 (0.5–1.6) |
| Female | 28 (46.7%) | Ref | Ref |
| **Age group** | | | |
| > = 18 years | 55 (91.7%) | 3.5 (1.5–10.0) | 18.0 (3.7–90.6) |
| 2–17 years | 5 (8.3%) | Ref | Ref |
| **Occupation** | | | |
| Farmer, manufacturing, fisherman | 33 (55.0%) | 3.0 (1.0–12.7) | 2.4 (0.7–10.9) |
| Transportation (e.g., taxi/bus driver) | 2 (3.3%) | 5.5 (0.6–39.5) | 6.2 (0.7–48.8) |
| Office worker | 1 (1.7%) | 0.4 (0.02–3.6) | 0.8 (0.04–7.7) |
| Student | 10 (16.7%) | 1.1 (0.3–5.1) | 4.3 (0.9–24.1) |
| Other | 11 (18.3%) | 0.9 (0.3–4.0) | 0.9 (0.2–4.0) |
| Homemaker | 3 (5.0%) | Ref | Ref |
| **Education** | | | |
| Less than primary school | 1 (1.7%) | 1.43(0.1–8.0) | 0.2 (0.01–1.6) |
| Primary school | 17 (28.3%) | 2.2 (0.9–5.5) | 0.4 (0.1–2.1) |
| High school | 32 (53.3%) | 2.1 (1.0–5.1) | 0.5 (0.1–2.6) |
| College | 2 (3.3%) | 0.4 (0.1–1.6) | 0.1 (0.01–0.9) |
| Currently in school | 8 (13.3%) | Ref | Ref |
| **Travel prior to onset** | | | |
| No | 59 (98.3%) | 5.4 (1.2–96.7) | 6.8 (1.1–138.3) |
| Yes | 1 (1.7%) | Ref | Ref |

*Other includes Laos (16), Venezuela (8), Mexico (1), Myanmar (1), and South Africa (1).

## Discussion

Our two-year sentinel surveillance project provided information on the etiology of patients presenting with acute febrile illnesses at five sentinel hospitals in Southern China. Ten different bacterial, viral, and protozoan etiologies were detected. Dengue was the most common. However, positive TAC results for *O. tsutsugamushi*, *C. burnetii*, and *Rickettsia spp.* are noteworthy, primarily due to known laboratory diagnostic challenges and under-reporting. These findings highlight the importance of routine testing for these infections to inform clinical management and the development of effective prevention and control measures. Despite the presence of competent mosquito vectors (e.g., *A. aegypti and/or albopictus*) in Mengla Country and Jiangmen City, Zika virus was not identified as a cause of AFI among patients enrolled in this project.

Our findings are consistent with AFI studies conducted elsewhere in South and Southeast Asia [17], which underscore the importance of dengue in the overall AFI disease burden, particularly following the decline of malaria transmission in the region. In Mengla County,

**Table 6. Association between patient characteristics/epidemiologic factors and detection of *Coxiella burnetii* on TaqMan Array Card (TAC) diagnostic testing platform, China, June 2017–July 2019.**

| Characteristic/Epidemiologic risk | TAC+ *C. burnetti* | Crude OR (95% CI) | ORa (95% CI) |
|---|---|---|---|
| | n (%) | | |
| **Location** | | | |
| Mengla County | 1 (2.4%) | 0.02 (0.001–0.09) | 0.02 (0.001–0.1) |
| Jiangmen City | 41 (97.6%) | Ref | Ref |
| **Nationality** | | | |
| Other* | 0 (0%) | - | - |
| Chinese | 42 (100%) | Ref | Ref |
| **Gender** | | | |
| Male | 8 (19.0%) | 3.5 (1.7–8.2) | 3.3 (1.5–8.2) |
| Female | 34 (81.0%) | Ref | Ref |
| **Age group** | | | |
| > = 18 years | 41 (97.6%) | 12.8 (2.8–228.9) | 2.4 (0.2–103.1) |
| 2–17 years | 1 (2.4%) | Ref | Ref |
| **Occupation** | | | |
| Farmer, manufacturing, fisherman | 7 (16.7%) | 0.2 (0.07–0.7) | 0.5 (0.1–1.6) |
| Transportation (e.g., taxi/bus driver) | 3 (7.1%) | 3.9 (0.7–18.6) | 2.7 (0.4–15.7) |
| Office worker | 3 (7.1%) | 0.6 (0.1–2.1) | 1.1 (0.2–5.4) |
| Student | 1 (2.4%) | 0.04 (0.002–0.2) | 0.1 (0.004–1.5) |
| Other | 21 (50.0%) | 0.8 (0.3–2.1) | 1.0 (0.4–1.5) |
| Homemaker | 7 (16.7%) | Ref | Ref |
| **Education** | | | |
| Less than primary school | 0 (0%) | - | - |
| Primary school | 13 (30.9%) | 13.6 (2.7–247.9) | 2.1 (0.1–87.1) |
| High school | 23 (54.8%) | 12.4 (2.6–223.4) | 1.0 (0.05–40.9) |
| College | 5 (11.9%) | 8.2 (1.3–158.0) | 0.6 (0.03–27.1) |
| Currently in school | 1 (2.4%) | Ref | Ref |
| **Travel prior to onset** | | | |
| No | 39 (92.9%) | 1.1 (0.4–4.7) | 1.5 (0.4–6.7) |
| Yes | 3 (7.1%) | Ref | Ref |

*Other includes Laos (16), Venezuela (8), Mexico (1), Myanmar (1), and South Africa (1).

repeated introductions of dengue from Laos and lack of local herd immunity were associated with the fall 2017 dengue outbreak, consistent with the epidemiology of the previous dengue outbreak in Yunnan Province in 2013 [19]. Despite consistent ecological risk factors (i.e., subtropical climate), the lower proportion of AFI cases with dengue positive TAC results in Jiangmen City could reflect a lower risk for imported cases or the impact of the intensive vector control policies and practices (against *Aedes* spp. mosquitos) implemented in the highly urban centers in Guangdong Province following the large dengue outbreak in the Pearl River Delta in 2013 to 2015 [30].

Approximately 13% of all AFI patients–and almost 30% of patients testing positive for at least one pathogen on TAC–were infected with *O. tsutsugamushi* or *C. burnetti*, pathogens responsible for scrub typhus and Q fever, respectively. In addition to undifferentiated *Rickettsia* spp., these infections are the cause of severe febrile morbidity and mortality worldwide but are often under-diagnosed and under-reported [31–33]. Scrub typhus is a nationally notifiable disease in China. A previous analysis of 93,481 scrub typhus cases reported between 2006 and

2016 identified high-risk locations in rural and recently developed areas in Yunnan and Guangdong Provinces [34], with human infections typically peaking in summer and autumn each year [33]. Patients with positive TAC results for *O. tsutsugamushi* in our project followed this seasonal pattern (consistent with the lifecycle of the mite, *Leptotrombidium delicense*, responsible for human infections in southern China).

Q fever is not a notifiable disease in China, and diagnostic testing is not widely available. *C. burnetti* infections can occur in people working with livestock and exposed to highly infectious aerosols from birth products, infectious dust particles or contaminated wool [35, 36]. Although human infections have been previously identified in China, primarily from seroprevalence studies [37], our project detected 41 acute Q fever cases in Jiangmen City. Most Q fever cases were male and employed in non-farming industries in Siqian Township and Enping County. These sex and occupational risks factors are consistent with the global epidemiology of Q fever [35]. Information on Q fever and scrub typhus was reviewed during a recent One Health Zoonotic Disease Prioritization workshop in Beijing [38]. Both diseases were ranked among the top 16 priority zoonotic diseases in China, highlighting the importance of a multi-sectoral approach in the detection, prevention, and control of pathogens responsible for AFI.

Due to the close economic and cultural connections between Yunnan and the upper Mekong River Basin, including Laos and Myanmar, we anticipated that the Zika virus would most likely be detected among AFI patients in Mengla County. Retrospective analysis of stored blood samples suggests that Zika virus had been circulating at low levels in Thailand [39], Laos [40], and Myanmar [41] since at least 2006. In 2016, 13 persons returning to Enping County from South America tested positive for Zika virus [24]. However, these persons were identified through two-month screening program at Baiyun International Airport in Guangzhou, with follow-up of travelers at their place of residence. Despite potential exposures, Zika virus was not detected during the two-year surveillance project. Zika virus infection is most concerning for pregnant women and the risk of microcephaly in the developing fetus [42]. During this project, three pregnant women had positive TAC detections (two for dengue and one for *C. burnetti*). To the best of our knowledge, no negative health impact on the patients or newborns resulted from these infections.

The project relied on standard case definitions, eligibility criteria, and TAC customized for Asia, addressing the primary methodological and reporting recommendations for AFI etiology investigations [11]. We detected 2 of the 13 viruses, 7 of 11 bacteria, and 1 of 3 protozoan targets included on TAC. The project also identified several discrepancies between TAC and malaria and dengue rapid diagnostic test results. Previous investigations on the use of the Wondfo RDT in field practice suggest that combing the malaria RDT with microscopy could increase both testing sensitivity and specificity [43]. Similarly, combining the NS1 antigen testing with serology-based assays for IgM and IgG may improve the accuracy of dengue diagnosis, particularly in areas with a lower prevalence of dengue transmission [44]. The large percentage of AFI patients hospitalized at the time of enrollment could reflect the clinical severity of the infection, project recruitment practices (e.g., more convenient blood draw), or presumed dengue diagnoses in which patients with dengue were admitted regardless of disease severity. The two-year project period suggested seasonality for AFI and several pathogens, which can increase clinical awareness and inform public health interventions.

Despite important findings on the etiology of AFI in Southern China, the two-year surveillance project was subject to a few limitations. First, not all patients presenting with AFI during the enrollment period were recruited to participate. Although Xinhua District is relatively urban, townships in Mengla County and in Enping County were more rural, resulting in additional travel time and costs to reach project hospitals. Patients with mild or even moderate symptoms may have elected to seek care at commune-level health care facilities or self-treat at

home [45]. Secondly, enrollment of pediatric AFI patients was lower than expected, possibly reflecting the large volume (5 ml) of whole blood required for TAC. We also excluded patients presenting with cough and diarrhea from project enrollment, although children testing positive for Rickettisia spp., scrub typhus, and Q fever can often present with these clinical signs and symptoms [32, 33]. As a result, certain pediatric AFI cases may have been inadvertently excluded from our surveillance project. Thirdly, less than half of enrolled AFI patients had a positive TAC detection. Future AFI surveillance projects could combine TAC with multiplex serology platforms such as the Luminex AFI panel. This combined approach could increase diagnostic sensitivity for patients presenting late in the disease course. Fourthly, controls were only enrolled from the sentinel hospital in Mengla County during the second project year (July 2018 to August 2019). Test results from control groups can be used to assess attributable fractions; that is, the proportion of cases with a specific positive TAC result who have symptomatic illness due to the detected pathogen. In this project, all enrolled controls tested negative for the 27 pathogens on TAC. Finally, we were unable to collect information on the health outcomes of the AFI patients enrolled in the project. Several of the detected etiologies, including dengue, malaria, and *O. tsutsugamshi* can have high mortality rates, particularly if diagnosed late. All patients received care and treatment according to the Chinese national guidelines.

## Conclusion

The two-year sentinel surveillance project identified the etiologies of AFI patients from five sentinel hospitals in Southern China using TAC, a multi-pathogen diagnostic platform. Although dengue was identified as the primary cause of AFI, the detection of *O. tsutusgamshi* and *C. burnetti* highlights the geographical variability in AFI etiology and importance of differential diagnosis. Improved rt-PCR based diagnostic testing for patients presenting with AFI is warranted, particularly for working-aged males engaged in farming, manufacturing, and fishing industries and persons returning from international travel. Approaches to reduce the risk of work-related exposures to mites and infectious animal products are also warranted. The project provided a framework for sentinel surveillance for AFI in China.

## Disclaimer

The findings and conclusions in this publication are those of the author(s) and do not necessarily represent the official position of the US Centers for Disease Control and Prevention or the China Center for Disease Control.

## Supporting information

**S1 File.**
(DOCX)

## Acknowledgments

We thank Dr. Xiaopeng Qi with China CDC for her expertise in developing and deploying the customized AFI-EDDC web-based data reporting platform. We also thank Dr. Hong Chen, lead for the China CDC Emerging Infections Diseases program, for her contributions towards the successful implementation of the two-year project in Mengla County and Jiangmen City. Finally, we extend tremendous appreciation to the public health and medical staff at the five sentinel hospitals who supported daily enrollment and specimen collection from AFI patients during the two-year project period, despite challenging workloads and competing health care priorities.

## Author Contributions

**Conceptualization:** Jeanette J. Rainey, Yuzhi Zhang, Shuyu Wu, Jie Liu, Eric Houpt, Barry Fields, Zhonghua Yang, Changwen Ke.

**Data curation:** Xiafang Guo, Lina Yi, Zhonghua Yang, Changwen Ke.

**Formal analysis:** Jeanette J. Rainey, Casey Siesel, Xiafang Guo, Lina Yi, Yuzhi Zhang, Shuyu Wu, Adam L. Cohen, Jie Liu, Eric Houpt, Barry Fields, Zhonghua Yang, Changwen Ke.

**Methodology:** Jeanette J. Rainey, Casey Siesel, Xiafang Guo, Lina Yi, Yuzhi Zhang, Shuyu Wu, Adam L. Cohen, Jie Liu, Eric Houpt, Barry Fields, Zhonghua Yang, Changwen Ke.

**Visualization:** Casey Siesel.

**Writing – original draft:** Jeanette J. Rainey, Adam L. Cohen.

**Writing – review & editing:** Casey Siesel, Xiafang Guo, Lina Yi, Yuzhi Zhang, Shuyu Wu, Adam L. Cohen, Jie Liu, Eric Houpt, Barry Fields, Zhonghua Yang, Changwen Ke.

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
