## [Decision Letter · Decision Letter 0]

29 Mar 2022

PONE-D-22-05533Etiology of acute febrile illnesses in Southern China: findings from a two-year sentinel surveillance project, 2017-2019PLOS ONE

Dear Dr. Rainey,

Thank you for submitting your manuscript to PLOS ONE. After careful consideration, we feel that it has merit but does not fully meet PLOS ONE’s publication criteria as it currently stands. Therefore, we invite you to submit a revised version of the manuscript that addresses the points raised during the review process. I have received the reviews of your manuscript and both reviewers agreed that this is informative and interesting work.  However, there are still a few concern that needs to be addressed, please see reviewers' insightful comments below.  In addition, it will be nice to have more information regarding the outcome of the patients, including duration of hospitalization, treatment, recovery or death, etc.  Specific comments:1. Fig.1 is a black box in the PDF file, please upload a file that is viewable.2. Line 324 - 325, please change "*Coxiella burnetii*" and "*Orientia tsutsugamushi*", to "*C. burnetii*" and "*O.*
*tsutsugamushi*"

3. Line 378, suggest changing "...detected co-infection." to "....co-infection."4. Line 424, "...in summer and summer and autumn..." This sentence is a bit confusing, please rephrase.

We look forward to receiving your revised manuscript.

Kind regards,

Baochuan Lin, Ph.D.

Academic Editor

PLOS ONE

Journal Requirements:

"This AFI sentinel surveillance project was funded by the U.S.-China Collaborative Program on Emerging and Re-emerging Infectious Diseases Cooperative Agreement #5U2GGH000961-04."

"The AFI sentinel surveillance project was funded by the U.S.-China Collaborative Program on Emerging and Re-emerging Infectious Diseases Cooperative Agreement #5U2GGH000961-04

Reviewers' comments:

Reviewer's Responses to Questions

**Comments to the Author**

1. Is the manuscript technically sound, and do the data support the conclusions?

Reviewer #1: Yes

Reviewer #2: Yes

2. Has the statistical analysis been performed appropriately and rigorously? 

Reviewer #1: N/A

Reviewer #2: Yes

3. Have the authors made all data underlying the findings in their manuscript fully available?

Reviewer #1: Yes

Reviewer #2: Yes

4. Is the manuscript presented in an intelligible fashion and written in standard English?

Reviewer #1: Yes

Reviewer #2: Yes

5. Review Comments to the Author

Reviewer #1: Overall

This is an informative and interesting work. It is important to have the local epidemiology data of etiologies of acute febrile illness. The manuscript is well written; however, if it is possible, the manuscript should be more concise.

Major issues.

-The manuscript should be shortened and more concise.

Minor issues.

-Authors defined AFI as fever within 7 days without targeted organ symptoms (e.g., cough, sore throat, diarrhea). However, diarrhea is presented as clinical presentations in Table 2.

-Mean day of fever onset should be presented.

- For patient treatment outcomes, hospitalization was described but not death. If there was any death, it should be reported the disease. If not, should mention that there is no death.

Reviewer #2: Sadly the article only deals with the etiologic agent of AFI, sadly no information regarding the outcome, duration of hospitalization for each positive agent group of patient, any antimicrobial was given, outcome of the patients studied: any serious outcome: death or none at all

6. PLOS authors have the option to publish the peer review history of their article (what does this mean?). If published, this will include your full peer review and any attached files.

Reviewer #1: No

Reviewer #2: No

---

## [Author Response · Author response to Decision Letter 0]

12 May 2022

Response to Reviewer

Academic Editor

More information regarding the outcome of the patients, including duration of hospitalization, treatment, recovery or detach, etc. 

We greatly appreciate this comment. The project was designed as a surveillance project, focused on describing the etiology of patients presenting with AFI at one of the five sentinel hospitals. Project staff collected information from AFI patients at the time of enrollment, with limited follow-up on hospital and non-hospitalized patients. Although we understand the importance of having information on case outcomes, our project was not designed to follow patients prospectively. We do not have the specific outcomes for each patient due to the project design. We propose a new project to enroll and prospectively follow patients with positive TAC detections in the future. 

1) Fig 1 is black box in the PDF file, please upload file that is viewable

Thank you for letting us know that Figure 1 was not viewable in the PDF version received. We have verified that the current version meets the requirements of PLOS ONE and is viewable in the PDF file. If there are any remaining concerns with the image, please let us know. 

2) Line 324 – 325, please change “Coxiella burnetti” and “Orientia tsutsugamushi” to “C. burnettia” and “O. tsutsugamushi” 

Thank you for identifying this error. We have updated Coxiella burnetti” and “Orientia tsutsugamushi” to “C. burnetti” and “O. tsutsugamushi”, as recommended. 

3) Line 378, suggest changing “…detected co-infection” to “…co-infection”.

We appreciate this comment and have updated Line 378 (Line 347-348 in the revised manuscript) to read “This includes two patients in Mengla County with DENV-O. tsutsugamushi co-infections”, as recommended. 

4) Line 424, “…in summer and summer and autumn…” This sentence is a bit confusing, please rephrase. 

Thank you for this feedback. We have revised Line 424 (Line 393-394 in the revised manuscript) to read “…in rural and recently developed areas in Yunnan and Guangdong Provinces, with human infections typically peaking in summer and autumn each year.” 

Additional requirements

1) Confirm manuscripts meets PLOS ONE’S style requirements 

We have reviewed PLOS ONE’S style requirements and confirm that the revised manuscript (and related figures) meet these requirements. 

2) Review and ensure reference list is complete and accurate

We have reviewed our references and confirm that the list is complete and accurate. Because we focused on shortening the length of manuscript, a few of the references have been deleted and/or cited in different sections of the manuscript. 

3) Remove funding reference from Acknowledgements and ensure is accurate to be included in Funding Statement

As required per the PLOS ONE style requirements, we have removed any funding information from the Acknowledgement section. We have verified the accuracy of the statement below – that can be added Funding Statement by the Journal Editors. 

"The AFI sentinel surveillance project was funded by the U.S.-China Collaborative Program on Emerging and Re-emerging Infectious Diseases Cooperative Agreement #5U2GGH000961-04. The funders had no role in study design, data collection and analysis, decision to publish, or preparation of the manuscript."

4) Data availability - We note that you have indicated that data from this study are available upon request. PLOS only allows data to be available upon request if there are legal or ethical restrictions on sharing data publicly. 

The authors of the manuscript do not have the legal authority to share the data with our submission of the manuscript. The data collected and analyzed as part of this project covered by the Data Sharing agreement established at the beginning of the project, consistent with the rules and regulations of the China Health Commission. Access to the de-identified dataset can be requested in a letter/email to Dr. Ke Changwen at Kechangwen@cdcp.org.con (for the AFI case data from Jiangmen City) and Dr. Zhonghua Yang at 18987903669 (for the AFI case and control data from Mengla County). 

5) Ensure Ethics Statement is included in the Methods Section of the manuscript. 

We have confirmed that the Ethics Statement is included in the Methods Section.

Reviewer #1 Comments

1) The manuscript should be shortened and more concise

Thank you for this comment and feedback. Our goal is to publish an easy to read and informative manuscript. As recommended, we have shortened the manuscript to the best of our ability to 4,225 words.

2) Authors defined AFI as fever within 7 days without targeted organ symptoms (e.g., cough, sore throat, diarrhea). However, diarrhea is presented as a clinical presentation in Table 2.

Thank you for noting this inconsistency. We have deleted the AFI cases reporting diarrhea and bloody diarrhea/vomiting and regenerated the descriptive and analytical results. The exclusion of these cases did not impact the overall findings and conclusion from the project. Please see Table 2 for the breakdown of signs/symptoms of the 836 cases that meet the eligibility criteria and were included in the final AFI surveillance dataset. 

3) Mean day of fever onset should be presented

We appreciate this comment. Unfortunately, we did not collect the date of fever onset needed to calculate the mean duration of fever at the time of enrollment. Each hospital requested whether fever onset was during the last 7 days. If yes, and the maximum temperature was >= 37.5 C, the patient was eligible for participation (assuming that all of criteria were met). This approach may have inadvertently enrolled patients who temperature started more than 7 days before contact with the sentinel hospital as well as patients who incorrectly reported an elevated temperature at home, we anticipate that the number of patients incorrectly enrolled are likely to very small. We also anticipate that the mean duration of fever prior to enrolment was likely to vary by distance from the sentinel hospital and the severity of the illness. 

4) For patient treatment outcomes, hospitalization was described but not deaths. If there was any death, it should be reported by disease. If not, should mention that that was no death.

Thank you for this comment. As mentioned above, the project was designed as a surveillance project, focused on describing the etiology and possible risk factors of patients presenting with AFI at one of the five sentinel hospitals. Project staff collected information from AFI patients at the time of enrollment. The project protocol did not include patient follow-up on hospital and non-hospitalized patients. Although we understand the importance of having information on case outcomes, our project was not designed to follow patients prospectively. As such, we do not have the specific outcomes for each patient due to the project design. We propose a separate project to enroll and prospectively follow patients with positive TAC detections in the future. 

Reviewer #2 Comments

1) The article only deaths with the etiologic agent of AFI, and no information regarding the outcome, duration of hospitalization for each positive agent group of patients, any antimicrobial was given, outcome of the patients studied, or any serious outcome, death or none at all. 

Thank you for this comment. The project was designed as a surveillance project, focused on describing the etiology and possible risk factors of patients presenting with AFI at one of the five sentinel hospitals. Project staff collected information from AFI patients at the time of enrollment. The project protocol did not include patient follow-up on hospital and non-hospitalized patients. Although we understand the importance of having information on case outcomes, our project was not designed to follow patients prospectively. As such, we do not have the specific outcomes for each patient due to the project design. We propose a separate project to enroll and prospectively follow patients with positive TAC detections in the future.

---

## [Decision Letter · Decision Letter 1]

14 Jun 2022

Etiology of acute febrile illnesses in Southern China: findings from a two-year sentinel surveillance project, 2017-2019

PONE-D-22-05533R1

Dear Dr. Rainey,

We’re pleased to inform you that your manuscript has been judged scientifically suitable for publication and will be formally accepted for publication once it meets all outstanding technical requirements.

Kind regards,

Baochuan Lin, Ph.D.

Academic Editor

PLOS ONE

Additional Editor Comments (optional):

Reviewers' comments:

Reviewer's Responses to Questions

**Comments to the Author**

1. If the authors have adequately addressed your comments raised in a previous round of review and you feel that this manuscript is now acceptable for publication, you may indicate that here to bypass the “Comments to the Author” section, enter your conflict of interest statement in the “Confidential to Editor” section, and submit your "Accept" recommendation.

Reviewer #1: All comments have been addressed

Reviewer #2: (No Response)

2. Is the manuscript technically sound, and do the data support the conclusions?

Reviewer #1: Yes

Reviewer #2: Yes

3. Has the statistical analysis been performed appropriately and rigorously? 

Reviewer #1: Yes

Reviewer #2: Yes

4. Have the authors made all data underlying the findings in their manuscript fully available?

Reviewer #1: Yes

Reviewer #2: Yes

5. Is the manuscript presented in an intelligible fashion and written in standard English?

Reviewer #1: Yes

Reviewer #2: Yes

6. Review Comments to the Author

Reviewer #1: (No Response)

Reviewer #2: This manuscript only deal with etiology, no other clinical relation to etiology .

7. PLOS authors have the option to publish the peer review history of their article (what does this mean?). If published, this will include your full peer review and any attached files.

Reviewer #1: No

Reviewer #2: No

---

## [Editor Report · Acceptance letter]

20 Jun 2022

PONE-D-22-05533R1 

Etiology of acute febrile illnesses in Southern China: findings from a two-year sentinel surveillance project, 2017-2019 

Dear Dr. Rainey:

I'm pleased to inform you that your manuscript has been deemed suitable for publication in PLOS ONE. Congratulations! Your manuscript is now with our production department. 

Kind regards, 

on behalf of

Dr. Baochuan Lin 

Academic Editor

PLOS ONE